# A Narrative Review on the Phytochemistry, Pharmacology and Therapeutic Potentials of *Clinacanthus nutans* (Burm. f.) Lindau Leaves as an Alternative Source of Future Medicine

**DOI:** 10.3390/molecules27010139

**Published:** 2021-12-27

**Authors:** Tan Yong Chia, Chee Yuen Gan, Vikneswaran Murugaiyah, Syed F. Hashmi, Tabinda Fatima, Lazhari Ibrahim, Mohammed H. Abdulla, Farhan Khashim Alswailmi, Edward James Johns, Ashfaq Ahmad

**Affiliations:** 1Analytical Biochemistry Research Centre (ABrC), Universiti Sains Malaysia, Lebuh Bukit Jambul 11900, Penang, Malaysia; tanyongchia@usm.my; 2Department of Pharmacology, School of Pharmaceutical Sciences, Universiti Sains Malaysia, Lebuh Bukit Jambul 11800, Penang, Malaysia; vicky@usm.my; 3College of Pharmacy, University of Rhode Island, 7 Green House Road, Kingston, RI 02881, USA; fayazhashmi84@uri.edu.usa; 4College of Pharmacy, University of Hafr Al Batin, Hafr Al Batin 39524, Saudi Arabia; tabinda@uhb.edu.sa (T.F.); fswelmi@uhb.edu.sa (F.K.A.); 5Department of Clinical and Diagnostic Radiology, College of Medical Technology, University of Tripoli, Tripoli P.O. Box 13275, Libya; lazharim63@yahoo.com; 6Department of Physiology, School of Medicine, University College of Cork, T12 K8AF Cork, Ireland; mdapharm78@yahoo.co.uk (M.H.A.); ej.johns@ucc.ie (E.J.J.)

**Keywords:** *C. nutans*, medicinal uses, phytochemistry, pharmacology, therapeutic potential

## Abstract

The application of natural products and supplements has expanded tremendously over the past few decades. *Clinacanthus nutans* (*C. nutans*), which is affiliated to the Acanthaceae family, has recently caught the interest of researchers from the countries of subtropical Asia due to its medicinal uses in alternative treatment for skin infection conditions due to insect bites, microorganism infections and cancer, as well as for health well-being. A number of bioactive compounds from this plant’s extract, namely phenolic compounds, sulphur containing compounds, sulphur containing glycosides compounds, terpens-tripenoids, terpens-phytosterols and chlorophyll-related compounds possess high antioxidant activities. This literature search yielded about one hundred articles which were then further documented, including the valuable data and findings obtained from all accessible electronic searches and library databases. The promising pharmacological activities from *C. nutans* leaves extract, including antioxidant, anti-cancer, anti-viral, anti-bacterial, anti-fungal, anti-venom, analgesic and anti-nociceptive properties were meticulously dissected. Moreover, the authors also discuss a few of the pharmacological aspect of *C. nutans* leaves extracts against anti-hyperlipidemia, vasorelaxation and renoprotective activities, which are seldom available from the previously discussed review papers. From the aspect of toxicological studies, controversial findings have been reported in both in-vitro and in-vivo experiments. Thus, further investigations on their phytochemical compounds and their mode of action showing pharmacological activities are required to fully grasp both traditional usage and their suitability for future drugs development. Data related to therapeutic activity and the constituents of *C. nutans* leaves were searched by using the search engines Google scholar, PubMed, Scopus and Science Direct, and accepting literature reported between 2010 to present. On the whole, this review paper compiles all the available contemporary data from this subtropical herb on its phytochemistry and pharmacological activities with a view towards garnering further interest in exploring its use in cardiovascular and renal diseases.

## 1. Introduction

The application of natural products and supplements has expanded tremendously over the past few decades due to economical and less adverse effects when compared to modern day medicines. Presently, more than 80% of people worldwide rely on them for some part of primary healthcare, especially in underdeveloped nations where drugs are usually pricey and unattainable, which encourages people to adopt traditional remedies. Starting from the late 1980s to recent years, tremendous investigations on herbal plants as alternative therapeutic agents have been used to treat a plethora of ailments due to their inexpensive cost and lower risk of side effects [1]. At present, 350,000 higher plants have been identified and, in relation to this number, only 8000 species are claimed to have medicinal properties [2]. Acanthaceae is one of the advanced and specialized families of 250 genera with approximately 2500 species providing effective traditional remedies against various health conditions [3].

*C. nutans*, which is affiliated to the Acanthaceae family, has recently caught the interest of researchers from the countries of subtropical Asia because of its medicinal uses. There are various vernacular names of this plant that exists different communities. In Malaysia and Brunei, *C. nutans* is recognized as “Sabahan snake grass” or “Belalaigajah”; “Dandanggendis” or “Ki tajam” in Indonesia; “Phayayo” or “Saledpangpontuamea” in Thailand and “You dun cao” or “Sha be she cai” in China (Table 1).

*C. nutans* is a scandent shrub with upright branches drooping that is over around 1–3 m tall. Its foliage usually appears as a stalked leaf with lanceolate-ovate, lanceolate to linear-lanceolate about 4–12 cm long by 1–4 cm wide. It has dull red to orange red flowers about 3.2 cm long with a green base borne in dense terminal racemes. The fruits are in the form of a capsule that is 2 cm long with short hair (Figure 1).

This plant can be propagated via seed or stem cutting [10]. The ethnobotanical uses of *C. nutans* are popular in Malaysia, Indonesia, Thailand and China, where this plant is commonly used in folk medicine to treat skin rashes, herpes simplex virus-induced lesions and insect or snake bites, as well as hyperuricemia, gout, urinary complications, diabetes, renal insufficiency, hyperlipidemia and various inflammatory conditions including strains and sprains injuries, hematoma, contusion and rheumatism. Pharmacological research revealed that this plant contains antioxidants and compounds with anti-cancer, anti-viral, anti-inflammatory, antidiarrheal, anti-diabetic and renoprotective activity, as shown in Figure 2. Ever since 2011, there has been a sudden surge in the usage of *C. nutans* in the folks of Southeast Asia following remarkable news regarding a patient recovered from the final stage of lymph node cancer from Taiping, Malaysia (https://myherbs2017.wordpress.com/category/clinacanthus-nutans, accessed on 15 August 2021). This plant is also utilized as a therapeutic option in menstrual pain, anemia and jaundice, and repairs bone fractures according to some traditional Chinese medicine; however, more attention is required to determine the dosage and form of the plant for which it can be used as medicine [11]. Although the studied data reported the beneficial effects of roots, stem and whole parts of *C. nutans,* the most frequently used part of the plant was the leaves decoction with water for ingestion and immersed in alcohol for tropical application [12].

Due to *C. nutans* popularity, a wide range of commercial products are formulated in the form of concentrated liquid beverages, tea, soap, essential oil drops, massage oil, ointments, concentrated balms, creams, lotions, capsules and powder [10] (Table 2).

Scientific data were further enriched by various review articles which explained the pharmacological significance of *C. nutans* [1]. The current review was designed to present a comprehensive summary of the pharmacological significance of *C. nutans* in different ailments and to highlight the possible therapeutic properties of *C. nutans* leaves in the treatment of cardiovascular and renal diseases. Furthermore, the current review will provide future directions of research and product development of this potential plant. For this purpose, research data and literature were collected from several computerized databases up to July 2021 as available in PubMed, Google Scholar, MEDLNE, NCBI, Web of Science, EMBASE, Cochrane Library, Clinical Trials.org, SciFinder and Scopus. Moreover, unpublished materials, such as conference papers, ethnobotanical textbooks and M.Sc. or Ph.D. dissertations were adapted.

## 2. Phytochemistry *C. nutans*

The phytochemical classes that are present in *C. nutans* leaves are sulphur containing compounds, sulphur containing glycoside compounds, phenolic compounds, terpenstripenoids and terpens phytosterols compounds [13]. The details of sub-phytochemical compounds are tabulated in (Table 3) and chemical structure is shown in Figure 3

The most widely used screening tests used to elucidate the chemical constituents of *C. nutans* leaves are thin layer chromatography (TLC) and Fourier transform infrared spectroscopy (FTIR) because these techniques are easy to conduct, time effective and of low cost. However, some other conscientious techniques such as high-performance liquid chromatography (HPLC), liquid chromatography mass spectrometry (LCMS) and gas chromatography mass spectrometry (GCMS) are employed to provide guidelines on the functional groups and classes of the chemical constituents that are present in this plant (Table 4). It is noteworthy to mention that the post-harvesting and preparation prior to the extraction procedures are very crucial. Heterogeneity in soil and climate, stages of maturity, geographical location, storage duration and solvent used during the extraction process directly influenced the quality and quantity of the phytonutrients of the leaves. It was reported that the phenolic content was 26% higher in younger leaves compared with the mature plant; in addition to that, mature leaves had lower phytochemicals, ascorbic acids and chlorophylls content compared to their younger counterparts. Moreover, prolonged storage of *C. nutans* leaves reduces the chlorophyll and total phenolic constituents from 25% to 50%, respectively. In 2015, Raya and his team had demonstrated that *C. nutans* leaves harvested at a younger stage had higher ascorbic acid content and the outcome of the study revealed that increasing the storage duration from one to four days led to a reduction in ascorbic content from the extract by 2.2-fold [29]. This suggests that fresh leaves must be used in the shortest possible time for optimum efficacy.

On the other hand, Chelyn and her team reported that, among the entirety of compounds identified, only shaftoside was present in all leaf samples regardless of geographical location from which the leaf samples were procured [35]. Since shaftoside is the stable flavonoid, such evidence demanded that shaftoside can be used as a chemical marker for *C. nutans* leaves. Additionally, the nutritional compositions of *C. nutans* leaves has also been extensively elucidated by [16] using mineral, vitamin and proximate analysis as tabulated in Table 5.

## 3. Pharmacological and Medicinal Properties

Many plants have abundant active secondary metabolites that exhibit certain pharmacological effects in humans, and the investigation of these phytochemical constituents in medicinal plants has caught the attention of researchers worldwide. This is due to that the isolated bioactive compounds have the greatest contribution in nutraceutical and pharmaceutical industries. It has also been recognized that *C. nutans* has several promising therapeutic potentials, and the Thai Ministry of Public Health had shortlisted this plant into the “Thai Herbal National Essential Drug List” as one of the medicinal plants for their public healthcare policy on anti-viral activity [36]. Moreover, a non-scientific and unpublished survey of ethnobotanical applications of medicinal plants has demonstrated that *C. nutans* rated amongst the top five most commonly used herbs for anti-diabetic, anti-hypertensive, anti-inflammatory and antioxidant properties in other sub-tropical countries such as in Malaysia, Brunei and Singapore. Other pharmacological activities such as anti-venom, anti-cancer, anti-bacterial, anti-fungal and anti-analgesic activities have also been reported [10,20,37].

### 3.1. Antioxidant and Anti-Cancer Properties

From the biological point of view, antioxidants are compounds which are capable of preventing damage by oxidants or free radicals while the products of the reaction between antioxidant and oxidant should not be toxic and not a branch of the radical reaction [38]. In addition to these, the half-life of an effective antioxidant must be long enough to counteract the oxidant. Thus, as a potential antioxidant, it must always remain in sufficient concentration especially during disease prevention circumstances [39]. *C. nutans* leaves possess diverse medicinal potential in conventional applications. A study reported by Nik Abd Rahman and his team investigated the antioxidant effects of *C. nutans* extracts using bone marrow smearing, clonogenic and splenocyte immunotype analysis with two different concentrations; 200 mg/kg and 1000 mg/kg methanolic leaf extracts in a 4 -T1 tumor-bearing mice model. They reported that methanol extract from *C. nutans* leaves at 200 mg/kg and 1000 mg/kg significantly attenuated the nitric oxide (NO) and malondialdehyde (MDA) levels in the blood. Similarly, *C. nutans* extract from leaves at 1000 mg/kg decreased the number of mitotic cells, tumour weight and tumour volume. From this study, no inflammatory or adverse reactions related to splenocytes activities were found in all treated groups of mice. Moreover, the concentration of both *C. nutans* leaf extracts has also reduced the number of carcinogenic colonies formed in the liver and lungs. This shows that *C. nutans* leaf extracts exerted an antioxidant activity in the 4-T1 mouse breast model [40]. Likewise, a study lead by [41] has used *C. nutans* leaves extracted with 80% methanol and further fractionated with n-hexane, dichloromethane, chloroform, *n*-butanol and aqueous residue ranging between 125 and 4000 µg/mL, whereas the total flavonoid content, total phenolic content and total antioxidant scavenging activity on breast cancer (Michigan Cancer Foundation-7 [MCF7]) and normal breast (Michigan Cancer Foundation-10A [MCF 10A]) cell lines were measured using the 2,2-diphenyl-1-picrylhydrazyl (DPPH) radical scavenging method and 2,2′-azino-bis(3-ethylbenzothiazoline-6-sulfonic acid (ABTS) radical cation decolourization assay. Based on the findings, the total phenolic content in *C. nutans* leaf extracts was higher than total flavonoid content. On the contrary, the n-hexane fraction had the lowest antioxidant activity; however, the crude fraction had the highest antioxidant activity according to the EC50 value. On the other hand, [42] also reported the anti-proliferative activity of *C. nutans* leaves extracts against the HeLa cell line, and the dichloromethane fraction had the lowest IC50 value of 70 µg/mL post 48 h incubation period; this indicated that the HeLa cell line, when exposed to the dichloromethane fraction, exhibited remarkable morphological features of apoptosis to the HeLa cancer cell line. On the contrary, [36] reported that the crude methanol extract of *C. nutans* leaves had the lowest scavenging activity as compared to ethyl acetate and *n*-butanol fractions of the methanol extract. This contradictory finding attested that the phytochemical content of *C. nutans* leaves is largely influenced by environmental conditions, i.e., variations in pH and nutrients in soil, temperature, humidity and water variability. Moreover, environmental factors which interact with the genetics of the *C. nutans* plants may lead to genetic variations that affect the phytochemical contents [43] (Table 6a).

### 3.2. Anti-Viral Properties

*C. nutans* leaves have long been utilized in Thailand as an alternative traditional medicine for the treatment of herpes simplex virus, varicella-zoster virus, mosquito borne virus and many more. In general, the modes of action of anti-viral properties from *C. nutans* leaves were demonstrated with three different stages of treatment, i.e., direct inactivation, pre-infection and post-infection methods. An experiment on anti-mosquitoes borne virus was demonstrated using dengue viruses serotypes-2 strain 16681 by immunofluorescence technique and reverse transcriptase polymerase chain reaction. Here, the dengue virus serotypes-2 was treated by incubating the dengue virus either in the absence or presence of the *C. nutans* leaf compounds in a sub-cytotoxic concentration at 37 °C for two days via pre-incubation and post-incubation techniques. The results showed that a phaeophorbide-a methyl ester compound identified in extracts could inhibit the dengue virus serotypes-2’s replication in a post-incubation study, which indicated that phaeophorbide-a could inhibit the production of viral RNA as well as viral the protein when the virus serotypes-2 infected cells were cultured in the compound [44]. In the treatment of herpes simplex virus (HSV), monogalactosyl diglyceride and digalactosyl diglyceride compound extracted from *C. nutans* leaves were tested using a plaque reduction assay method for their in-vitro anti-viral activities against herpes simplex virus type 1 and type 2. The result had demonstrated that the monogalactosyl diglyceride and digalactosyl diglyceride compounds which were present in the *C. nutans* leaves inhibited the replication of HSV type 1 post step of infection by 100% at non-cytotoxic concentration with IC_50_ values of 36.00 and 40.00 mg/mL, whereas the herpes simplex virus type 2 was at 41.00 and 43.20 mg/mL, respectively. This finding illustrated the inhibitory activity of *C. nutans* leaves extract against both herpes simplex virus serotypes could be probably via the inhibition of the late stage of viral multiplication, suggesting their promising use as anti-HSV agents [46]. The anti-papillomavirus infectivity of *C. nutans* leaves was evaluated using human papillomavirus 16 PsVs infection on the 293FT cell line. Based on the in-vitro study, DMSO and heparin extract of *C. nutans* leaves showed a potential anti-human papillomavirus 16 PsV infections effect by preventing the early step of infection between the direct bindings of human papillomavirus particles to the host cell receptor, while also preventing human papillomavirus 16 PsVs internalization [46]. On the other hand, a clinical evaluation of the anti-vera zoster virus in an aphthous stomatitis experiment was reported by [47], where a double blind controlled trial was undertaken to evaluate the efficacy of orabase *C. nutans* leaves extract in recurrent aphthous stomatitis patients. Patients were subjected to topical formulation of *C. nutans* leaves extract in the ulcers site and it was found that application or a base four times a day successfully reduced pain score and healed the Vera zoster virus lesion. The findings suggest the potential role of the *C. nutans* leaf compounds on the prevention of human papillomavirus infection and Vera Zoster virus infections (Table 6b).

### 3.3. Anti-Bacterial Properties

Antimicrobial resistance is a global health and development threat in the current century which requires urgent multi-sectorial actions in order to achieve Sustainable Development Goals. A lack of clean water, sanitation and inadequate infection prevention control further promotes the spread of microbes in some of poor countries, some of which can be resistant to antimicrobial treatment. Moreover, misuse and overuse of anti-microbials are the main drivers in the development of drug-resistant pathogens. For example, the rate of resistance to ciprofloxacin, an antibiotic commonly used to treat urinary tract infections, varied from 8.4% to 92.9% for *Escherichia coli* and from 4.1% to 79.4% for *Klebsiella pneumoniae* in those countries reporting to the Global Antimicrobial Resistance and Use Surveillance System. With the rise of these phenomena, scientists have changed focus to natural compounds in medicinal plants to identify potential new anti-bacterial compounds and, hence, the anti-bacterial effects of *C. nutans* leaves have been tested in microbial strains. Lim and his team have reported that the extracts from non-polar and polar *C. nutans* leaf extracts showed growth inhibition in all 12 bacteria species: *Bacillus subtilis, Enterobacter, Escherichia coli, Enterobacter aerogenes, Enterococcus faecalis, Klebsiella pneumoniae, Proteus vulgaris, Pseudomonas aeruginosa, Staphylococcus aureus, Staphylococcus epidermidis* and *Staphylococcus saprophyticus*; as the extracts concentration increased, the results revealed that non-polar *C. nutans* leaf extracts have a stronger antibacterial activity than those polar extract solutions at 32 mg/kg concentration, whereas the gram-negative bacteria were more sensitive to the extracts compared to gram-positive bacteria [48]. On the other hand, [49] reported that purpurin-18-phytyl-ester compound extracted from *C. nutans* leaves possesses in-vitro anti-biofilm wound healing activities in RAW 264.7 or the HGFs cell line. In addition to that, the anti-bacterial properties from ethanolic and chloroform fraction of *C. nutans leaves* were also reported against *Porphyromonas gingivalis* and *Aggregatibacter actinomycet emcomitans* using disc diffusion agar, minimum inhibitory concentrations (MIC) and minimum bactericidal concentrations (MBC) antibacterial susceptibility tests done in-vitro. Fifty percent ethanolic *C. nutans* leaves extract was found to have a notable antibacterial activity against *Porphyromonas gingivalis* and *Aggregatibacter actinomycetemcomitans*, comparable to 0.2% chlorhexidine. Meanwhile, chloroform *C. nutans* leaves extract was found to have notable anti-bacterial activity against *Porphyromonas gingivalis* only [50]. On the whole, this multiplicity of findings suggested that the anti-bacterial effects from *C. nutans* leaves extract could be selective for only particular strains of microorganisms, and, thus, the exact mode of action of *C. nutans* leaves extract on bactericidal effects still requires further extensive investigations and re-definition (Table 6c).

### 3.4. Anti-Fungal Properties

*C. nutans* leaves extract has been widely employed as a traditional medicine for anti-fungal activity in the countries of Southeast Asia. However, to date, there are somewhat limited scientific data available to support the claims that have been made, yet there are still some research findings proven to have positive results. For instance, Choon and his team investigated the inhibitory activity of aqueous *C. nutans* leaves extract against *Candida albicans* using agar disk diffusion and the micro-broth dilution technique. The result obtained showed negative inhibitory activity against *Candida albicans* [51]. The same finding was further supported by [52], who had examined the anti-fungal activity in *Candida albicans* and *Aspergillus fumigatus* with 95% ethanol leaf extract at 5 mg/mL. On the contrary, [22] has reported that a minimal concentration of 1.39 mg/mL of ethyl acetate extract exhibited a fraction of an antifungal effect on *Candida albicans*. Based on the above mixed findings, the polar and non-polar extract exhibited unpromising fungicidal action. There is still substantial room to explore the biological action of *C. nutans* leaves on anti-fungal activities, which is worthy of further investigation (Table 6d).

### 3.5. Anti-Venom Properties

Folks have always recognized that one kind of herb can be universally used for relieving symptoms from the venom of many animals or insect species. However, evidence has showed that venom can be neutralized by the body’s defense mechanism with disregard to any effects from herb treatment, which indirectly caused the misunderstanding of this plant as an antidote. Since the major elements that are present in venom are peptides and proteins with very delicate structure, pH or other uncomplicated factors can exert any effect, leading to confusion of their actions. In fact, data from the traditional healers are obvious. However, they prefer to keep this knowledge with themselves for their own profit and tend to end up with their data always lost without any records after they are deceased [54]. Generally, the extracts from *C. nutans* leaves are used by native and local people from Southeast Asia as the remedy for the envenomation of bites or stings by venomous animals or insects, i.e., snakes, scorpions and bees. Extracts are commonly prepared via direct maceration or using water, with ethanol as an extraction solvent. Previous in-vivo investigations have demonstrated contradictory result indicating that 95% alcoholic *C. nutans* leaves extract at a concentration of 0.406 mg/mL to 0.706 mg/mL was not sufficient to exert the antidote effect against the neurotoxin disseminated by *Naja naja siamensis* in isolated pherenic-nerve diaphragm preparations in rats [53]. Similarly, 0.406 mg/mL to 0.706 mg/mL of aqueous ethanolic *C. nutans* leaves extract was also ineffective against *Apis mellifera* Linn. In bee’s venom, the viability of fibroblast cell was less than 10% [53]. On the contrary, water extract was able to reduce the mortality rate against the neurotoxin from *Naja naja siamensis* by 27% [53]. Others have reported effective inhibitory potential of *C. nutans* leaves extract at a 1:12.5 dilution ratio against the *Naja naja siamensis* cobra venom using a modified ELISA technique with only 35% of inhibitory activity, indicating that the extract attenuated toxin activity by extending the contraction time of the diaphragm muscle after envenomation and had a potency to protect cellular proteins from venom degradative enzymes [55]. Likewise, an in-vitro study on the effectiveness of water *C. nutans* leaves extract successfully exhibited 46.5% fibro-blast cell lysis against Heterometrus laoticus scorpion venom at a 0.706 mg/mL concentration [56]. Based on the current available ambivalent results, advanced scientific efforts are necessary to clarify these plant activities (Table 6e).

### 3.6. Analgesic and Anti-Nociceptive Properties

Pain medication can be defined widely as any medication that relieves pain. Many different pain medicines exist and each has both pros and cons due to the fact that certain pains respond better to some medicines while some do not. Each individual also has a slightly different response to different pain relievers. Currently, the most common medications are over-the-counter medicines such as the non-steroidal anti-inflammatory drugs (NSAIDS) class, which are used for mild to moderate pain and are commonly prescribed for arthritis and musculoskeletal physiotherapy; the opioids class, which includes codeine, morphine and tramadol, are often prescribed for acute pain caused by traumatic injury, such as post-surgery neuropathic pain; anti-epileptic drugs such as pregabalin, gaberpentin and carbamazepine are used for chronic pain, i.e., neuropathic pain; anti-depressants such as amitriptyline and duloxetine are used for chronic pain, i.e., fibromyalgia. These medications often come with some unpleasant side effects, i.e., all NSAIDs come with the risk of gastrointestinal ulceration and bleeding; opioid analgesics commonly cause drowsiness, dizziness and respiratory depression; anti-epileptic drugs cause dizziness, drowsiness and swelling of the lower extremities including dry mouth, difficulty urinating, blurred vision and constipation. Other possible side effects of anti-epileptic drugs include hypotension, tachycardia, palpitations, weight gain and fatigue. The analgesic capabilities of methanolic leaves extract of *C. nutans* have been investigated to assess their comparative analgesic and muscle relaxant activities in a study conducted on BALB/c mice using gold and silver nanoparticles as the vehicle at a concentration of 50, 100 and 150 mg/kg per body weight and methanolic extract at a concentration of 100, 200 and 400 mg/kg per body weight included under a twisted wire traction technique for the muscle relaxant study, and the analgesic study was assessed by writhing (extension of hind limb, turning of trunk, and contraction of abdomen) that took place during the coming 10 min after treating with acetic acid. The muscle relaxant studies displayed that methanolic leaves extract of *C. nutans* encoated with silver nanoparticles were comparatively more efficient than methanolic leaves extract of *C. nutans* encoated with gold nanoparticles in a traction examination. Additionally, the analgesic studies exhibited that those gold nanoparticles, silver nanoparticles and methanolic extracts alone exhibited the maximum percentage reduction in acetic acid induced writhing at the concentrations of 50, 100 and 150 mg/kg per body weight by 48.02, 64.30 and 74.44%; 45.23, 60.00 and 71.50%; 42.30, 58.00 and 69.33% writhing at 100 mg/kg, 200 mg/kg and 400 mg/kg, respectively. These findings indicated that *C. nutans* leaves extract possesses very good analgesic and muscle relaxant activities for use in pain management. On the other hand, [33] have demonstrated peripherally and centrally mediated anti-nociceptive activity via the modulation of the opioid/NO-mediated pathway using sequentially partitioned to obtain petroleum ether extract from *C. nutans* leaves, which was subjected to an anti-nociceptive study with the aim of establishing its anti-nociceptive potential by determining the role of opioid receptors and L–arginine/nitric oxide/cyclic-guanosine monophosphate (L-arg/NO/cGMP) pathway in the observed anti-nociceptive activity. In the study, 100, 250 and 500 mg/kg of petroleum ether extract from *C. nutans* leaves were orally administered and the abdominal constriction, hot plate and formalin-induced paw licking test in mice was investigated. In addition to that, the effect of petroleum ether extract from *C. nutans* leaves on locomotors activity was also evaluated using the Rota-rod assay. The test outcome showed that petroleum ether extract from *C. nutans* leaves significantly inhibited the nociceptive effect in all models in a dose-dependent manner; except that the highest dose of petroleum ether extract from *C. nutans* leaves, 500 mg/kg, did not affect the locomotors activity of treated mice. The authors concluded that the anti-nociceptive activity of petroleum ether extract from *C. nutans* leaves significantly inhibited all antagonists of μ-, δ- and κ-opioid receptors. In addition, the anti-nociceptive activity of petroleum ether extract from *C. nutans* leaves was reversed by L-arg, but was somehow insignificantly affected by morphine hydrochloride. This result suggested that petroleum ether extract from *C. nutans* leaves could exert an anti-nociceptive activity at peripheral and central levels possibly via the activation of nonselective opioid receptors and modulation of the NO-mediated partly via the synergistic action of phenolic compounds presence in the plant extracts (Table 6f).

### 3.7. Anti-Inflammatory and Immunomodulatory Properties

Anti-inflammatory agents block certain substances in the body that cause inflammation and are used to treat many different disease conditions. Some anti-inflammatory agents are being studied in the prevention and treatment of cancer. On the contrary, an immunomodulatory substance suppresses or stimulates the immune system that helps the body to fight against cancer, infection, or other diseases. Specific immunomodulating agents, i.e., monoclonal antibodies, cytokines and vaccines, affect specific parts of the immune system. Extracts from *C. nutans* leaves have been adopted to reduce inflammation in viral infection, insect bites and allergic responses in medicine. A few investigations have also reported the effect of *C. nutans* leaves extract on the immune system. The anti-inflammatory study was assessed by in-vitro assays such as on interleukin-4 (IL-4) and interleukin-13 (IL-13) cytokines secretion in phorbol-12-myristate-13-acetate (PMA)-induced U937 macrophage cells. In this study, a sequential ultrasonic-assisted extraction was carried out using water and ethanol, with a 1:10 ratio of leaves powder to the solvent volume at 0.25, 0.5, 1.0, 2.0, 4.0 and 8.0 mg/mL concentration. Viability of the extract-treated cells using the Presto-Blue test and the IL-4 and IL-13 secretions were assessed with the ELISA technique which caused morphological changes in U937 cells from round-shaped, non-adherent to larger irregular-shaped, adherent cells, and a reduction in cells viability to 87%. Moreover, the CD14 expression was down-regulated by 36% upon PMA stimulation together with the CD11b expression being up-regulated by 58% in PMA-treated cells. ELISA results showed that 1 mg/mL of ethanolic and water extracts stimulated 1200 and 1800 pg/mL IL-4 secretions, respectively, but both extracts caused minimal IL-13 secretion which indicates that aqueous extracts stimulated IL-4 production higher than ethanolic extract in PMA-induced U937 macrophages, suggesting that inflammatory effects could be dampened with such doses [57]. It also reported that 80% ethanolic extract from leaves showed 68.33% inhibition on the generation of superoxide anion and the elastase release by activated neutrophils in 10 µg/mL ethanolic extract. MeO-Suc-Ala-Ala-Pro Valp-nitroanilide was used for observing elastase release and superoxide anion production by detecting the superoxide dismutase-inhibitory reduction from ferricytochorme c complex. On the other hand, the immunomodulating study examined the inhibitory effect of *Lactobacillus casei* on IgE production, splenocyte obtained from ovalbumin (OVA)-primed BALB/c mice and re-stimulated in-vitro with the same antigen. In this immune-modulating experiment, administration of 0.1 μg/mL of 80% ethanol extract led to up-regulation of IFN-γ exhibiting immune-modulating activity [58] (Table 6g).

### 3.8. Anti-Hyperglycemic Properties

Large numbers of studies have provided evidence for the significant role of oxidative stress in diabetes, obesity and some form of metabolic syndromes. Oxidative stress occurs due to an imbalance between endogenous antioxidant systems and the generation of reactive oxygen species (ROS). ROS overproduction has been reported to be an important trigger of insulin resistance and a contributing factor in the development of type-2 diabetes [66]. Diabetes is a chronic disease that occurs when the pancreas does not produce enough insulin. The common effect of uncontrolled diabetes over time leads to serious damage to many of the body’s systems, especially the nerves and blood vessels which are responsible for the development of cardiovascular disease, with approximately 80% of cardiovascular mortality and morbidity linked to vascular complications. According to the statistical report from the World Health Organization, the number of personnel with diabetes rose from 108 million in 1980 to 422 million in 2014. Prevalence has been rising more rapidly in low- and middle-income countries than in high-income countries. At present, it has been estimated that up to one-third of personnel suffering from diabetes mellitus adopted some form of ethnomedical applications. One of the medicinal plants that caught the attention of diabetic patients for its perceived anti-diabetic properties is *C. nutans*. The anti-hyperglycemic effect was demonstrated via aqueous leaf extract on serum metabolic indices, sorbitol production and aldose reductase enzyme activities in the kidneys, ocular lens and sciatic nerve of type-2 diabetic (T2D) rats at a concentration of 100 and 200 mg/kg/day p.o., potentially lowering the fasting blood glucose levels post-intervention by 14.2 and 14.0 mmol/L, respectively, at week four, compared with the untreated group 22.1 mmol/L. In addition to that, *C. nutans* leaves extract also attenuated the oxidative stress marker, namely F_2_-isoprostane, with an enhancement of aldose reductase enzyme activity increased by 64 and 99%. These findings indicated that *C. nutans* leaves extract has the potential to attenuate type-2 diabetic-induced metabolic perturbations and complications [59]. Moreover, [60] have also reported that 500 mg/kg/daily of *C. nutans* leaves extract reverts endothelial dysfunction in type-2 diabetes rats by improving protein expression of endothelial nitric oxide synthase (eNOS) enzyme with respect to 300 mg/kg/daily of metformin. Treatment of both diabetic groups with *C. nutans* leaves extract or metformin improved the impairment of endothelium-dependent vasorelaxation associated with up-regulated expression of aortic eNOS protein. Moreover, *C. nutans* leaves extract and metformin also reduced aortic endothelium-dependent and aortic endothelium-independent contractions in diabetic rats. Both of these diabetic-treated groups had reduced blood glucose levels and increased body weight compared to the untreated diabetic group. This finding indicated that *C. nutans* leaves extract could be a potential anti-diabetic therapy in the future as it displayed a similar therapeutic outcome as compared to metformin. The anti-diabetic potential of *C. nutans* leaves extract was also studied in-silico via the characterization of α-glucosidase inhibitors by gas chromatography-mass spectrometry-based metabolomics and molecular docking simulation using 80% methanolic dried leave samples. GC-MS data analysis discovered 11 bioactive compounds including palmitic acid, phytol, hexadecanoic acid, 1-monopalmitin, stigmast-5-ene, pentadecanoic acid, heptadecanoic acid, 1-linolenoylglycerol, glycerol monostearate, alpha-tocospiro B and stigmasterol. Some of the potential inhibitor compounds were identified from the leaves extract and the molecular interactions of the inhibitors identified with the protein were predominantly hydrogen bonding-involving residues, namely LYS156, THR310, PRO312, LEU313, GLU411, ASN415, PHE314 and ARG315 residues with hydrophobic interaction. This finding supported scientific evidences of the potential of *C. nutans* leaves in α-glucosidase enzyme inhibition, ideal for the development either on medicinal preparations, nutraceutical and novel therapeutic or preventive agents for future anti-diabetic treatment [61] (Table 6h).

### 3.9. Anti-Hyperlipidemia Properties

Globally, there are now more people who are obese, and this trend is observed in every region over the world. It is suggested that, by the year of 2030, the population experiencing overweight and obesity in adults will reach 2.16 billion worldwide [67]. Obesity can be defined as abnormal or excessive fat accumulation that may impair the body’s health state and elevated body mass index is a major risk factor for many non-communicable diseases such as: increases cardiovascular risk factors via increased fasting plasma triglycerides, elevated low density lipoprotein levels cholesterol, lowered high density lipoprotein cholesterol, elevated blood glucose and insulin levels and high blood pressure [68]. One of the plants with medicinal properties that includes crude extracts and isolated compounds which are effective for controlling and reducing weight gain is from *C. nutans* leaves. Treatment of high fat diet induced obese mice with methanolic leaf extract of *C. nutans* at 500, 1000 and 1500 mg/kg for 21 days reduced the body weight gained, visceral fat and muscle saturated fatty acid compositions. Moreover, the levels of HSL, PPAR α and PPAR γ and SCD gene expressions in the obese mice treated with 1500 mg/kg methanolic leaf extract of *C. nutans* were downregulated [62]. A similar finding was also reported where 39.0 and 58.5 mg/mL of methanolic leaves extract significantly lowered the area, size, and diameter of adipocyte. Although supplementation of *C. nutans* methanolic leaf extract could reduce plasma total cholesterol in mice, it was somehow not effective on other plasma lipid profile regulations [63]. In addition to that, *C. nutans* was able to slow down the rate of weight gain induced by high fat-high cholesterol diet in insulin resistance in rats, and also improved the antioxidant capacity in the obese rats. This anti-hyperlipidemic effect was mediated by the up-regulation of gene coding for phosphatidylinositol-3-phosphate, insulin receptor substrate, leptin and adiponectin receptors [16] (Table 6i).

### 3.10. Vasorelaxation Properties

As stated in Hypertension Clinical Practice Guidelines of Malaysia, thiazide, diuretics, β-blockers, CCBs, ACEIs and ARBs were selected as first line mono-therapeutic agents; however, several adverse side effects such as dizziness, fatigue, joint pain and stomach upset, constipation, dehydration, erectile dysfunction and low effectivity were always being reported. For this reason, the discovery of the new drugs to control blood pressure enchanted a number of researchers. Previously, *C. nutans* leaves showed limited data reports as antihypertensive agents. The prevalence of diabetes, dyslipidemia and hypertension are always responsible for a substantial risk of cardiovascular diseases. Reduced nitric oxide bioavailability may lead to endothelial dysfunction and hypertension which is thought to be related to loss of eNOS cofactor such as tetrahydrobiopterin, further substantiating oxidative stress to induce vascular pathogenesis [69]. It has been reported that extracts from *C. nutans* leaves contain several active ingredients which can undergo multiple vasorelaxation-mediated signaling pathways and decrease the time taken to achieve the targeted blood pressure with less concomitant adverse effects. Reference [11] has demonstrated a preliminary test to screen for their antihypertensive and vasorelaxant activities of *C. nutans* leaves using Fourier transform IR (FTIR), second-derivative IR (SD-IR) and two-dimensional correlation IR (2D-correlation IR) analyses to determine the main constituents and the fingerprints from this herb. In addition to that, water extracts, 50% ethanol extracts and 90% ethanol extracts from *C. nutans* leaves were used to determine the contractile forces on the pre-contracted aortic rings measured with a GRASS Force-Displacement Transducer FT03C on adult male Sprague–Dawley rats. Based on their findings, the vasorelaxant activities were prominent with the highest R_max_ values of 95% ethanol extracts (72.67 ± 1.61%) vs. 50% ethanol extracts (73.57 ± 2.99%) vs. water extracts (55.85 ± 2.35%). This outcome revealed that the flavonoid content obtained from this herb possesses a potential vasorelaxant activity (Table 6j).

### 3.11. Renoprotective Properties

*C. nutans* has also been evaluated for its renoprotective activities. The renoprotective effect of *C. nutans* has been demonstrated by several in-vivo and in-vitro studies [63,64,65]. In 2017, the nephroprotective effect of *C. nutans* leaves against cisplatin-induced nephrotoxicity and the safety assessment of *C. nutans* leaves has been demonstrated by [63]. The study has demonstrated that cisplatin-induced renal toxicity caused rapid loss of glomerular filtration, polyuria, hyperkalemia, hypernatremia and azotemia in animal models. Their protective activities on renal tubular cells (NRK52-E) were evaluated for cellular viability (MTT assay) and apoptosis (Hoechst and Rhodamine 123 staining). In vivo studies of *C. nutans* leaves were administered via oral gavage at doses of 100, 200 or 400 mg/kg for 90 days, while receiving weekly doses of cisplatin (1 mg/kg). Simultaneous treatment with cisplatin and *C. nutans* leaves extract significantly attenuated the renal toxicity manifested by decreased levels of serum creatinine and proteins, blood urea nitrogen, urine electrolytes and urine volume when compared to the cisplatin group. Furthermore, an increase in the glomerular filtration rate, serum electrolytes and urine creatinine excretion were demonstrated. Collectively, these findings highlighted the potential use of *C. nutans* leaves extract in the management and treatment of cisplatin-induced nephrotoxicity. Meanwhile, [64] has also reported the nephroprotective effect of *C. nutans* in cisplatin-induced nephrotoxicity under an in-vitro condition using the Proton Nuclear Magnetic Resonance (^1^H NMR) and Liquid Chromatography Mass Spectroscopy (LCMS) techniques coupled with multivariate data analysis to characterize the metabolic variations in intracellular metabolites and the compositional changes of the corresponding culture media in rat renal proximal tubular cells (NRK-52E). Investigations of this study have highlighted the altered pathways perturbed by cisplatin induced nephrotoxic on NRK-52E cells which involved changes in amino acid metabolism, lipid metabolism and glycolysis such that choline, creatinine, phosphocholine, valine, acetic acid, phenylalanine, leucine, glutamic acid, threonine, uridine and proline as the main metabolites which differentiated the cisplatin induced group of NRK-52E from control cells extract while the corresponding media exhibited lactic acid, glutamine, glutamic acid and glucose-1-phosphate as the varied metabolites. *C. nutans* aqueous leaves extract at 1000 μg/mL exhibited the highest potential for a nephroprotective effect against cisplatin toxicity on NRK-52E cell lines at 89% of viability where the protective effect of *C. nutans* aqueous leaves extract could be discerned by the changes in the metabolites such as choline, alanine and valine in the pre-treated samples with those of the cisplatin-induced group [64]. Moreover, the nephroprotective effect of *C. nutans* against cisplatin-induced nephrotoxic human kidney cells was also reported by the same group using 8 different solvent extracts from *C. nutans* leaves. The aqueous extract showed a protective effect against the induced cell line based on the improvement of the percentage viability in mitochondrial dehydrogenase activity (MTT) and lactate dehydrogenase (LDH) assay pretreated with the extract after 24 h [65] (Table 6k). Renoprotective data by *C. nutans* showed that many areas need to be unfolded and extensive research is required for bench work to clinical practice.

### 3.12. Toxicology Studies

Toxicity testing provides the knowledge regarding some of the risks that may be associated with use of herbs, therefore avoiding potential harmful effects when used for therapeutic purposes. Generally, toxicological studies could be divided into acute, sub-acute and chronic phases, depending on the exposure duration of animals to any drugs but also depending on the dose of the substance and also on the toxic properties of the substance. The relationships between these two factors are crucial in the evaluation of therapeutic dosage in pharmacology and herbalism [70]. In an acute in-vivo study, rats treated with 5000 mg/kg survived throughout the 14-day observation period. Neither death nor signs of toxicity-related changes were reported on skin and fur, eyes, mucous membranes, respiratory pattern, autonomic or behavior patterns such as convulsions, salivation, diarrhea or lethargy including changes in the body weight, water and food consumption in animals. The authors also reported that single dose administration of aqueous *C. nutans* leaves extract over 14 days showed no early or late morbidity, mortality or apparent signs of toxicity [unpublished data]. As in hematological parameters, the serum eosinophil level in rats treated with 500 mg/kg of aqueous *C. nutans* leaves extract was elevated by 1.9 times as compared to their control counterparts; however, the authors claimed that the variation was within the normal physiological range (0 < 2.5 < 3%) for rats. In addition to that, rats treated with 2000 mg/kg aqueous *C. nutans* leaves extract for 90 days showed increases in the activated partial thromboplastin time by 3.7 times compared to the normal control group. Their finding suggested that 2000 mg/kg of aqueous *C. nutans* leaves extract could potentially act as an anti-inflammatory therapeutic and anticoagulant agent [71]. In another study by [72], acute oral toxicity of methanolic extracts treated to male Swiss albino mice at 900 and 1800 mg/kg for 14 days did not exhibit any mortality cases and side effects on kidney, liver, lungs, spleen and heart. While, for sub-chronic toxicity study, the no-observed-adverse-effect level (NOAEL) is greater than 2500 mg/kg/day but its renal creatinine level was elevated at doses of 500 and 2500 mg/kg/day in a sub-chronic toxicity study [15]. In a study reported by [73], 1.3 g/kg of ethanolic *C. nutans* leaves extract administered subcutaneously, intraperitoneally and orally did not produce any signs of acute toxicity in rats. On the contrary, a recent study had claimed that sub-acute administration of the extracts once at 2000 mg/kg induced mild hepatic and renal histological alterations in mice. Similarly, repeated daily oral administration of *C. nutans* leaves extract for 28 days induced mild to moderate hepatic degeneration at 500 mg/kg and renal necrosis at 1000 mg/kg in female ICR mice [74,75]. These polarized findings could be due to insufficient scientific studies being conducted previously; moreover, the majority of those experiments were done as preliminary and fundamentally oriented. Therefore, further sophisticated investigation still needs to be done, due to lack of data obtained from biological investigations that are associated with other phytochemical bioactive fractions from this plant still leading to toxicity incidents in laboratory animals. This further pinpoint that the phytochemical compositions that are present in this plant extract used for toxicity studies were not fully identified via any phytochemical analysis in order to compare the biological studies. Farsi also simulated the use of this extract on a human equivalent dose, based on the results obtained from oral toxicological studies using the body surface area (BSA) normalization method, illustrating the human equivalent dose of aqueous *C. nutans* leaves extract is equal to 324.32 mg/kg [63]. However, the acceptable daily intake on the non-observable adverse effect level value obtained from the animal study is 9 mg/kg in humans with reference to the guidelines from Food and Agriculture Material Inspection Center [10,76]. Hence, a well-designed clinical study is still needed to assess its chronic toxicity to affirm a specific safety dose to be adopted for human consumption to avoid any potential adverse side effects.

## 4. Conclusions and Future Directions

*C. nutans* plant extracts are richly endowed with anti-oxidant, anti-cancer, anti-viral, anti-bacterial, anti-fungal, anti-venom, analgesic and anti-nociceptive, anti-hyperlipidemia, vasorelaxation and renoprotective activities. *C. nutans* leaves extract is a suitable source for alternative medicine in human; however, it is a medicine with low toxicity.

Based on the extensive laboratory investigations either in-vivo or in-vitro, it was proven that *C. nutans* leaves extract possesses a variety of phytochemical properties which have a wide range of therapeutic activities against several diseases. Although scientific studies have shown that this plant extract is richly endowed with anti-oxidant, anti-cancer, anti-viral, anti-bacterial, anti-fungal, anti-venom, analgesic and anti-nociceptive, anti-hyperlipidemia, vasorelaxation and renoprotective activities, at present, research data that focus on the pharmacological activities of *C. nutans* leaves extract against vasorelaxation activity were not abundant. Alternative lines for future studies could include the investigation of anti-hypertensive activity such as that in L-NAME and Doca-salt induced hypertensive model; the pathophysiology closely mimicked their human analogs. Despite the renoprotective properties of *C. nutans* leaves extract being reported, current available data were restricted to the cisplatin induced renal insufficiency model. The renoprotective potential of this plant could be further extended to other renal disease models such as cyclosporine or gentamicin induced renal failure model or the 2K1C and 1K1C animal models, which could greatly contribute to the knowledge of cardiovascular disease studies since renovascular hypertension is increasingly related to the pathogenesis of chronic kidney diseases. A second important unmet need is to resolve ongoing controversies concerning the insufficient previous scientific research data which were conducted due to most of their studies being preliminary and fundamentally oriented. Therefore, to facilitate future application of *C. nutans* leaves extract in clinical settings, a more sophisticated assessment pathway that analyses the aforementioned biological and its therapeutic efficacy are urged prior to the implementation in pharmaceutical sectors. Moreover, the in-vitro studies did not fully mimic the real physiological environment in humans. Therefore, additional in-vivo studies are to be done in cell line and further clinical trials, together with more experimental studies, are of the utmost important to substantiate the correlation of the isolated phytochemical compounds with their corresponding pharmacological effects in order to fully illustrate the effect of *C. nutans* leaves extract on disease prevention.

## Figures and Tables

**Figure 1 molecules-27-00139-f001:**
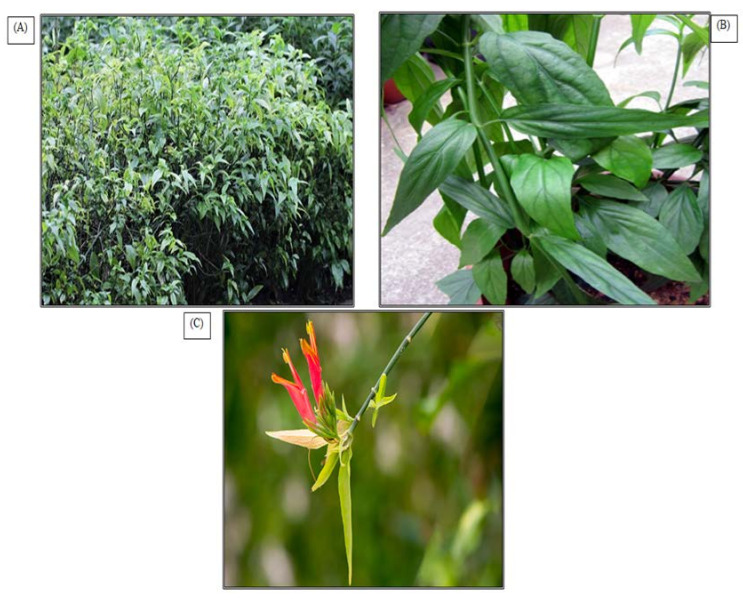
*C. nutans* in (**A**) plant habitat, (**B**) stems and leaves and (**C**) inflorescence (photographs of the plants were taken directly by using Canon, model number 3611C011, Tokyo, Japan).

**Figure 2 molecules-27-00139-f002:**
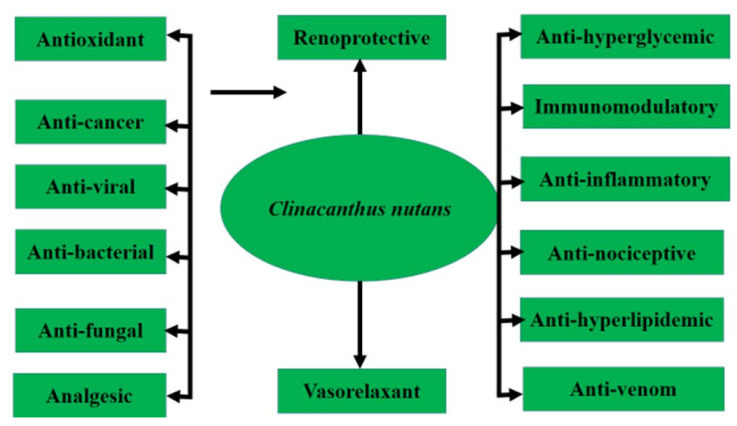
Pharmacological activities of *C. nutans*.

**Figure 3 molecules-27-00139-f003:**
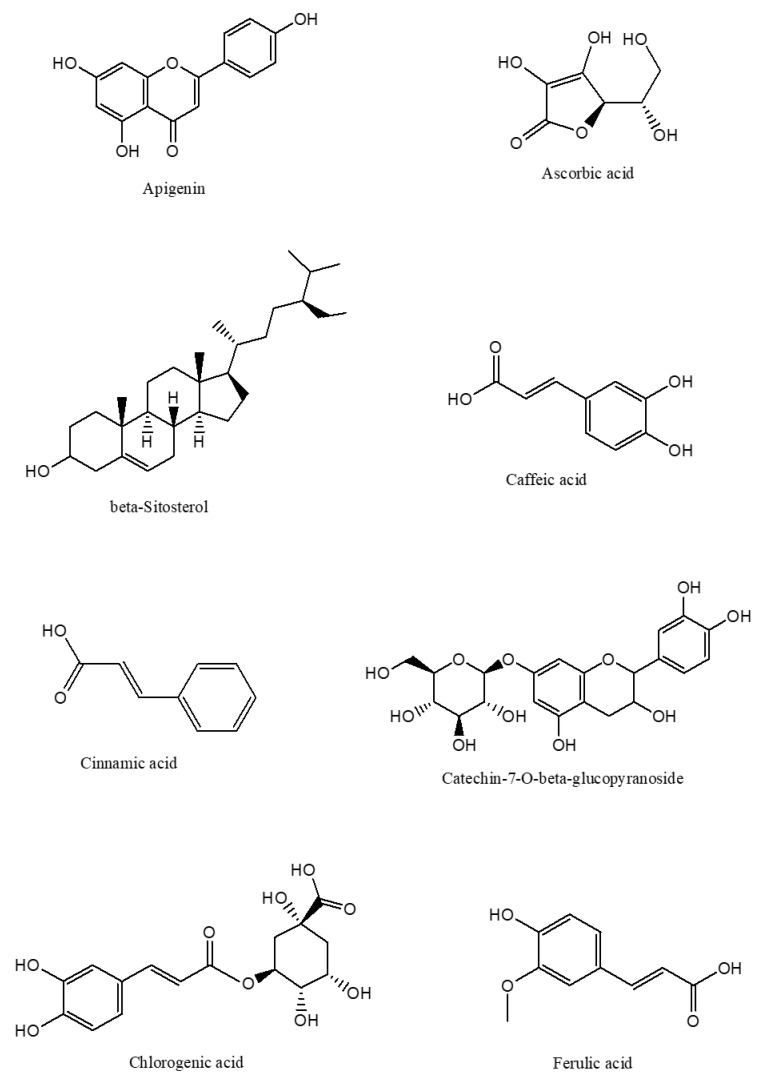
2D-chemical structure from various phytochemical classes and compounds present in *C. nutans* leaves (diagram was reconstructed using Chem-Draw ultra-Structure Software).

**Table 1 molecules-27-00139-t001:** Common vernacular names of *C. nutans*.

Country	Language	Vernacular Names	References
Malaysia, Brunei	Malay, English	Pokok stawa ular, Belalai gajah, Sabahan snake grass	[3]
Thailand	Thai	Phaya Plongtong, Phaya Yo, Saled Pangpon Tua Mea	[4,5]
China	Mandarin	E zui hua, Sha be she cao, You dun cao,	[5,6,7]
Indonesia	Jawa	Ki tajam Kijatan Daun dandang gendis	[3,8,9]

**Table 2 molecules-27-00139-t002:** A wide variety of commercial products made from *C. nutans* leaves for traditional and modern uses.

Formulation	Therapeutic Purposes
Tea	Anti-cancer, anti-diabetic, anti-hypertensive and body detoxification
Essential oil drop	Relieves oral herpes viral infection and aphthous ulcer
Soap or body wash	Treatment of skin problems and blemishes
Cream	Treatment of Herpes zoster and Herpes genitalis infection
Lotion	Relieve urticarial, itching and rashes
Powder	Anti-cancer, anti-hypertensive and anti-diabetic
Ointment	Relieve aches, cramps, sprains of muscular and joint, cold, flu and insect bites
Balm	Relieve insect bites, skin rashes, inflammation, muscular pain and dizziness
Elixir	Anti-cancer, alleviate period pain and diuretic properties for urinary tract and kidney diseases
Capsules	General health maintenance, body detoxification, anti-diabetic and anti-hypertensive

Adapted from: [10].

**Table 3 molecules-27-00139-t003:** The phytochemical classes and compounds present in *C. nutans* leaves.

Phytochemical Class	Phytochemical Compound	References
Phenolic compounds	3,3-di-o-methylellagic acid, 4-vinylphenol, 4-Hydroxybenzoic acid, 6,8-di-c-α-l-arabinopyranoside, 7-hydroxyflavone, 7-o-β-glucopytanoside, Acorbic acid, Apigenin, Apigenin-6-c-β-d-glucopyranosrl-8-c-α-l- arabiopyranoside, Caffeic acid, Cinnamic acid, Chlorogenic acid, Ferulic acid, Gallic acid, Gendarucin A, Isomollupentin, Isoorientin, Isovitexin, Kamferol, Orientin, Oxoprolinates Protocatechuic acid, Quercetin, Rutin trihydrate, Shaftoside, Syringic acid, Vanillin, Vitexin, Vanillic acid	[3,14,15,16,17,18]
Sulphur containing compounds	2-*cis*-entadamide A, Clinamides A, B, C, D & E, Entadamine A & C	[19,20,21,22]
Sulphur containing glycosides compounds	Clinacoside A, B & C, Cycloclinacoside A1 & A2	[19,23]
Terpens-tripenoids	Lupeol	[19,24]
Terpens-phytosterols	β-sitosterol, Stigmasterol, Stigmasterol-β-d-glucoside Stigmasteryl-3-o-β-glucopyranoside	[3,25,26,27]
Chlorophyll related compounds	13^2^-hydroxyl-(13^2^-*R*)-chlorophyll b, 13^2^-hydroxyl-(13^2^-*S*)-chlorophyll b, 13^2^-hydroxyl-(13^2^-*R*)-phaeophytin a, 13^2^-hydroxyl-(13^2^-*R*)-phaeophytin b, 13^2^-hydroxyl-(13^2^-*S*)-phaeophytin a	[5,28]

**Table 4 molecules-27-00139-t004:** The phytochemical functional group obtained from the screening of *C. nutans* leaves.

Extraction Solvent	Functional Group	References
Water	Alkaloid, Diterpenes, Phytosterol, Saponin, Triterpenoids	[30,31,32,33]
70% methanol	Carbohydrate, Flavonoids, Glycosides, Phytosterol, Protein and amino acid, Steroids, Tannin	[30,31,34]
100% methanol	Flavonoids, Phenolic compound, Phytosterol, Saponin, Steroids, Triterpenoids	[27,31,34]
100% chloroform	Alkaloid, Flavonoids, Glycosides, Tannin	[30,31,34]

**Table 5 molecules-27-00139-t005:** Nutritional composition of *C. nutans* leaves.

Nutritional Composition	Percentage (%)	Weight (mg/100 g)
Ash	10.0 ± 0.20	
Calcium		874.50 ± 31.25
Carbohydrate	73.27 ± 3.14	
Copper		0.26 ± 0.01
Fat	0.50 ± 0.02; 2.11 ± 0.66	
Fiber	2.71 ± 0.05	
Moisture	9.28 ± 0.40; 78.30 ± 0.29	
Potassium		1097.90 ± 6.93
Protein	5.16 ± 0.08; 5.73 ± 0.14	
Sodium		6.78 ± 1.01
Vitamin B1	0.27 ± 0.04	
Vitamin C	1.57 ± 0.07	

Adapted from [16]

**Table 6 molecules-27-00139-t006:** Pharmacological effect of *C. nutans* leaves.

Pharmacological Activity	Extract/Fraction	Dose Tested/ Test Method	Animals/ Cell line Culture (*In Vivo/In Vitro*)	Experimental Model/ Clinical Trial	Result	References
**(a) Antioxidant and Anti-cancer Properties**						
Antioxidant, protection against oxidative stress and anti-tumor	80% methanolic leaves extracts	200 mg/kg and 1000 mg/kg methanolic extracts	Murine mammary carcinoma cell line, 4-T1 cells (In-vitro); Female BALB/c mice (In-vivo)	Antitumor and antioxidant in 4-T1 tumor bearing mice	Significant decrease in NO and MDA levels in the blood. High dose (1000 mg/kg) extracts significant decrease the number of mitotic cells, tumor weight, and tumor volume	[40]
Antioxidant scavenging activity, and anti-proliferative effects on breast cancer cells	80% methanolic leaves extracts, and further fractionated sequentially with different solvents (hexane, dichloromethane, chloroform, *n*-butanol, and aqueous residue)	MCF 10A cells started with 500 µg/mL (CN-crude and CN-aqueous) and 120 µg/mL (CN-hexane, CN-dichloromethane, CN-chloroform, and CN-butanol fraction extracts	Breast cancer (Michigan Cancer Foundation-7 [MC ]) and normal breast (Michigan Cancer Foundation-10A [MCF 10A]) cells (In-vitro)	Molecular docking simulation of the major compounds from *C. nutans* leaves extract was conducted	Total phenolic content of *C. nutans* leaves extract was higher than that of total flavonoid content. CN-dichloromethane extract had the strongest anti-proliferative effect thatinhibited MC cell growth and less toxic towards MCF 10A cells	[41]
Anti-proliferative activity of extracts of *C. nutans* leaves against human cervical cancer (HeLa) cells	80% methanol or water extract. The methanol extract was further extracted to obtain hexane, dichloromethane and aqueous fraction	4,000 µg/mL (water, 80% methanol, and its aqueous fraction) or 250 µg/mL (hexane and dichloromethane fractions of the methanol extract	HeLa cells (ATCC^®^CCL-2™) (In-vitro)	HeLa cells using the Sulforhodamine B (SRB) assay	Extracts wereanti-proliferative against HeLa cells, and the dichloromethane fraction had the lowest IC_50_ value of 70 µg/mL at 48 h. Microscopic studies showed that HeLa cells exposed to the DCM fraction exhibited marked morphological features of apoptosis	[41]
Antioxidant and α-glucosidase inhibitory activity, with a subsequent analysis of total phenolic and total flavonoid content of methanol extract	Liquid-liquid partition chromatography in a separating funnel using hexane, methanol, and water (13:2:5)	Total phenolic content determined spectrophotometrically using Folin-Ciocalteu method, total flavonoid content estimated according to the based on the formation of aluminum-flavonoid complexes. DPPH for free radical scavenging capacity and FRAP method for total antioxidant capacity	Antioxidant and α-glucosidase inhibitory activities of methanol extract and its different fractions from *C. nutans* leaves using biochemical assays (In–vitro)	Identified the various chemical constituents of the extract and fractions by GC Q-TOF MS, in addition to bioactivity correlation	Ethyl acetate and butanol fractions of the methanol extract had the highest antioxidant and α-glucosidase inhibitory activity which showed a significant correlation with the total phenolic and total flavonoid contents of the fractions	[36]
**(b) Anti-viral Properties**						
Study compounds from *C. nutans* leaves on dengue virus type 2 infection	hexane and chloroform leaves extract	20 µL of leaves extract in MTT (3-(4, 5-dimethylthiazol-2-yl)-2, 5-diphenyl tetrazolium bromide. Incubate 3 h at 37 °C	A549 cell monolayers grown in 24- well tissue culture plates were adsorbed with the 0.01 MOI of treated dengue virus serotype-2 (In-vitro)	Anti-viral activity in pre-incubation vs. post-incubation period and tested using ELISA and RT-PCR	Phaeophorbide- a methyl ester compound was identified in the extracts could inhibit the dengue virus serotypes-2 replication in post-incubation study	[44]
Anti-herpes simplex virus activities of monogalactosyl diglyceride and digalactosyl diglyceride from *C. nutans leaves*	chloroform leavesextract	100 mL of Vero cells at concentration 2.5 × 10^5^ cell/mL seeded into culture medium at 37 °C with 5% CO_2_ for 1 day with differentconcentrations of chloroform crude extract (20 mL) using MTT assay	Vero cells (African green monkey kidney cells) cultured with Dulbecco’s modified Eagle medium, supplemented with 5% fetal bovine serum. (In-vitro)	Cytotoxicity of viral activities was evaluated by the MTT assay samples were measured at 490 nm via spectrophotometry	100% inhibition of herpes simplex virus type 1 replication at the post step of infection with IC_50_ values of 36.00 and 40.00 mg/mL, and herpes simplex virus type 2 at 41.00 and 43.20 mg/mL respectively	[45]
Anti-papillomavirus infectivity of *C. nutans* compounds	136B, 136C and 136D of *C. nutans* compounds dissolve in DMSO and heparin solution	The amount of viable cells was determined by adding 20 µL of 5 mg/mL MTT solution using 293FT cells dissolved with 100 µL of DMSO using an ELISA reader	Human Papillomavirus 16 PsVs co-transfection of P16 shell and pfwB into 293FT cells (In-vitro)	Human Papillomavirus 16 PsVs treated with or without various concentrations of each compound. Human papilloma virus 16 PsVs were adsorbed directly on 293FT cells, infected cells expressing green fluorescent protein and determined under fluorescent microscope	136B, 136C and 136 D compounds inhibited the early step of infection by direct binding between human papillomavirus particles and host cell receptor and also prevent human papillomavirus 16 PsVs internalization.	[46]
Anti-Vera zoster virus infection in oral ulcer	Topical formulation	4 times daily on infected area and assessed at least 3 times during treatment course	Human oral cavity (In-vivo)	Recurrent aphthous stomatitis	Reduces pain score and healed the lesion caused by Vera zoster virus	[47]
**(c) Anti-bacterial Properties**						
Growth inhibition in all twelve bacteria species: *Bacillus subtilis, Enterobacter, Escherichia coli, Enterobacter aerogenes, Enterococcus faecalis, Klebsiella pneumoniae, Proteus vulgaris, Pseudomonas aeruginosa, Staphylococcus aureus, Staphylococcus epidermidis, Staphylococcus saprophyticus*	Non-polar and polar *C. nutans* leaves extract	*C. nutans* leaves extract range from 0.25, 0.5, 1, 2, 4, 8, 16, 32 mg/mL	Cell lines in triplicate containing 40µL of bacterial suspension (final conc. of 2–8 × 10^5^ cfu/mL) with 50 µL of test compound inoculated with 10 µL of resazurin. (In-vitro)	Broth micro dilution method was used to determine the minimum inhibitory concentration (MIC) based on the CLSI M07-A8 guidelines	Growth inhibition in all 12 bacteria species as extract concentration increased. Non-polar extracts have stronger antibacterial activity than those polar extracts solution in the 32 mg/kg concentration	[48]
Anti-biofilm, nitric oxide inhibition and wound healing potential of purpurin-18-phytyl ester (P18PE) isolated from *C. nutans* leaves	*C. nutans* leaves -hexane (20.05 g), chloroform (16.2 g) and ethanol (38.34 g) extracts	*C. nutans* leaves extract range from (5–500 μg/mL)	1 × 105 cells (RAW 264.7 or HGFs)/well were incubated overnight in 96-well plates.	Murine macrophage RAW 264.7 and HGF cell culture.	Possess anti-inflammatory, in-vitro wound healing, and anti-biofilm activities.	[49]
Antibacterial properties of *C. nutans leaves* extracts against *Porphyromonas gingivalis* and *Aggregatibacter actinomycetemcomitans*	100%, 50%, 10% ethanol and 100% chloroformextracts	*C. nutans* leaves extract range from 12.5, 25, 50 and 100 mg/kg	Disc diffusion agar, minimum inhibitory concentrations (MIC), and minimum bactericidal concentrations (MBC) antibacterial susceptibility tests (In-vitro)	Disc diffusion agar test	50% ethanolic extracts have notable antibacterial activity against *P. gingivalis* and *A. actinomycetamcomitans* comparable to 0.2% chlorhexidine. Meanwhile, chloroform extract has notable antibacterial activity against *P. gingivalis* only	[50]
**(d) Anti-fungal Properties**						
*In-vitro* Anti-Candida Effect of Thai Herbs Supplemented in Tissue Conditioner	10% aqueous extracts	*C. nutans* leaves range from 0.354, 0.709, 1.418, 2.836, 5.672, 11.344 µL/mL) in tissue conditioner	Agar disk diffusion (inhibition zone appearance) and micro-broth dilution (MIC and MFC determination Methods (In-vitro)	Liquid part of COE-COMFORTTM tissue conditioner	Negative inhibitory activity against *Candida albicans*	[51]
Light-mediated activities against *Candida albicans* and *Aspergillus fumigatus*	95% ethanolic extracts	*C. nutans* leaves extract at 5 mg/mL	Agar disk diffusion (In-vitro)	Disc diffusion agar test	Extracts were ineffective to exhibit fungicidal effect on both fungus species	[52]
In-vitro anti-fungal activities of *C. nutans* leaves extract and semi-fractions	Crude extracts (0.2 to 10.0 mg/mL) subjected to cold solvent extraction to produce petroleum ether, ethyl acetate and methanol crude extracts, followed by isolation using bioassay-guided fractionation.	*C. nutans* leaves extract at 2 mg/mL, 4 mg/mL, 6 mg/mL, 8 mg/mL, 10 mg/mL	HeLa and K-562 cell lines cultured in RPMI1640 and DMEM complete medium (In-vitro)	Fungal suspensions were streaked on MHA and SDA medium followed by 3-(4,5-dimethylthiazol2-yl)-2,5-diphenyltetrazolium bromide (MTT), minimum inhibitory concentration (MIC) and minimum fungicidal (MFC) assay	A minimal concentration of 1.39 mg/mL of ethyl acetate extract exhibited a fraction of antifungal effect on *Candida albicans*	[22]
**(e) Anti-venom Properties**						
Extracts of *C. nutans* and *Naja naja siamensis* venom	95% alcoholic *C. nutans* leaves extract	0.406 mg/mL to 0.706 mg/mL administered orally or intraperitoneally	Mice (In-vivo)	Isolated rat phrenic-nerve diaphragm in mice	Failed to exert the antidote effect against the neurotoxin	[53]
*C. nutans* extract activities against the fibroblast cell lysis	0, 50 or 90% ethanolic extracts	0.406 mg/mL to 0.706 mg/mL	Chick embryonic fibroblast cell primary cultures (In-vitro)	Swiss Webster female mice	Completely negative results as anti-bee venom agents	[54]
Extracts of *C. nutans* Burm. and Naja naja siamensis venom	Water extract	0.406 mg/mL to 0.706 mg/mL administered orally or intraperitoneally	Mice (In-vivo)	Isolated rat phrenic-nerve diaphragm in mice	Reduced mortality rate by 27%; from 100% to 63 ± 3.34%	[53]
Screening of *C. nutans* containing *Naja naja siamensis* cobra venom inhibitory activity using modified ELISA technique.	Water extract	extracts at 1:250 pre-incubated for 30 min at 37 °C with 40 g/mL venom in DMEM (test) or without DMEM lacking venom(control)	Modified ELISA technique (In-vitro)	Phrenic nerve/ Hemi-diaphragms isolated from adult albino rats, weighing 150 to 200 g	35% of inhibitory activity and the extract attenuated toxin activity by extending contraction time of diaphragm muscle	[55]
Screening of *C. nutans* plant acting against *Heterometrus laoticus* scorpion venom activity on fibroblast cell lysis	Water extract	0.406 mg/mL to 0.706 mg/mL pre-incubated with DMEM (as mock controls), or with 0.2 g/L venom	Chick embryonic fibroblast cell primary cultures (In-vitro)	Chick embryonic fibroblast cell	Exhibited 46.5% fibroblast cell lysis in *Heterometrus laoticus* scorpion venom at 0.706 mg/mL but its cytotoxic effect is unsure	[56]
**(f) Analgesic and Anti-nociceptive Properties**						
*C. nutans* leaves extract mediated silver and, gold nanoparticles on muscle relaxant, analgesic activities for pain management	Methanolic extract encoated in silver and gold nanoparticles	50, 100, 200 mg/kg per body weight in gold and silver nanoparticles; 100, 200, 400 mg/kg per body weight in methanol extract	Intra-peritoneal injection of extracts on BALB/c mice (In-vivo)	Twisted wire traction technique for muscle relaxant study and writhing for analgesic study	Extract exerted a very good analgesic and muscle relaxant activities for use in pain management. Gold nanoparticles had most efficient analgesic activity at a small concentration of 50 mg/kg	[paper retracted]
Anti-nociceptive activity of petroleum ether fraction obtained from methanolic extract of *C. nutans leaves* via opioid receptors and NO mediated/cGMP-independent pathway	Petroleum ether fraction from methanolic extract of *C. nutans leaves*	100, 250, 500 mg/kg administered intraperitoneally	Adult male ICR mice (In-vivo)	Acetic acid-induced abdominal constriction test, hot plate test, formalin–induced paw licking test, and motor coordination Rota–rod test	Petroleum ether *C. nutans* leaves extract exerted anti-nociceptive activity at peripheral and central levels via the activation of nonselective opioid receptors	[33]
**(g) Anti-inflammatory and Immunomodulatory Properties**						
Effects of *C. nutans* leaves extract on cytokine secretion in PMA-induced U937 macrophage cells	Water and ethanol leaves extract	0.25, 0.5, 1.0, 2.0, 4.0 and 8.0 mg/mL	Viability of the extract-treated cells using Presto-Blue test; IL-4 and IL-13 secretion tested via ELISA (In-vitro)	U937 monocyte-derived macrophages	*In-vitro* assays on interleukin-4 (IL-4) and interleukin-13 (IL-13) cytokines secretion in PMA-induced U937 macrophage cells showed reduction of cells viability to 87%, CD14 expression was down-regulated by 36% and CD11b expression was up-regulated by 58%.	[57]
Anti-Inflammatory and immune-modulating activity in *C. nutans* leaves extract	80% ethanol leaves extract	0.1 to 10 µg/mL ethanolic extract	Anti-inflammatory: MeO-Suc-Ala-Ala-Pro Valp-nitroanilide was used for observing elastase release and superoxide anion production; Immune-modulating: *Lactobacillus casei* on IgE production, splenocyte obtained from ovalbumin (OVA)-primed BALB/c mice (In-vivo)	Ovalbumin (OVA)-primed BALB/c mice	68.33% inhibition on the generation of superoxide anion and the elastase release by activated neutrophils by 10 µg/mL ethanolic extract; 0.1 μg/mL of 80% ethanol extract led to up-regulation of IFN-γ	[58]
**(h) Anti-hyperglycemic Properties**						
Aqueous leaf extract of *C. nutans* improved metabolic indices and sorbitol-related complications in type II diabetic rats	Hot water extraction method where leaves are mixed with water in a 1:10 ratio (*w/v*) for 3 h at 100 °C	100, 200 mg/kg/day of water extract	Male Sprague Dawley rats (In-vivo)	Streptozotocin induced diabetic rat model	Improved glycemic control and complications. In fact at higher doses (200 mg/kg), *C. nutans* leaves extract showed better results	[59]
*C. nutans* leaves extract reverts endothelial dysfunction in type-2 diabetes rats by improving protein expression of eNOS	Methanolic extract from leaves	300, 500 mg/kg/day of methanolic extract	Male Sprague Dawley rats (In-vivo)	Intraperitoneal injection of low-dose streptozotocin to rats fed with high-fat diet	Improved endothelium-dependent relaxation, reduced endothelium -dependent and endothelium-independent contraction in the aorta of diabetic rats	[60]
Characterization of α-glucosidase inhibitors from *C. nutans* Leaves by Gas Chromatography-Mass Spectrometry-based metabolomics and molecular docking simulation	80% methanol using the sample to solvent ratio of 1:3 (*w/v*) for 3 days where the solvent was changed each consecutive day. i.e., hexane; hexane: ethyl acetate; ethyl acetate; ethyl acetate: methanol	10 µL from each sample extracts	gas chromatography-mass spectrometrybased metabolomics and molecular docking simulation (In-silico)	α-glucosidase inhibitory potential of *C. nutans* using the gas chromatography tandem with mass spectrometry (GC-MS)	α-glucosidase inhibitors were identified in *C. nutans* leaves, indicating the plant’s therapeutic effect to manage hyperglycemia	[61]
**(i) Anti-hyperlipidemia properties**						
Effects of methanolic leaf extract of *C. nutans* on fatty acid composition and gene expression in male obese mice	Methanolic leaves extract	500, 1000, 1500 mg/kg of leaves extract	Male ICR mice (In-vivo)	High fat diet induced obesity mice	Reduced the body weight, visceral fat and muscle saturated fatty acid compositions and down-regulated the levels of HSL, PPAR α and PPAR γ and SCD gene expressions with 1500 mg/kg had optimum efficacy	[62]
Methanolic Extract of *C. nutans* Leaves can alter adipocyte Cellularity activity in male obese mice	Methanolic leaves extract	19.5, 39.0 and 58.5 mg/mL of leaves extract	Male ICR mice (In-vivo)	High fat diet induced obesity mice	Lowered adipocyte area, size, and diameter and reduced plasma total cholesterol in mice but had no effect on plasma lipid profile	[63]
Effects of phenolic-rich extracts of *C. nutans* on high fat and high cholesterol diet-induced insulin resistance	Water and 80% aqueous methanol leaves extract	oral gavage of 125, 250 or 500 mg/kg/day of leaves extract	Male Sprague-Dawley rats (In-vivo)	High fat and high cholesterol rat	Slowed the rate of weight gain induced by high fat-high cholesterol diet	[16]
**(j) Vasorelaxation Properties**						
Anti-hypertensive and vasodilatory effects *C. nutans* leaves extract	Water extract, 50% ethanol extract and 95% ethanol extract from leaves	100 µL of herbal extracts added cumulatively to the organ bath at concentrations from 0.125 mg/mL to 128 mg/mL (equivalent to 0.00125–1.28 mg/mL in organ bath)	Male Sprague-Dawley rats (In-vitro)	Pre-contracted aortic rings from rat thoracic aorta	Prominent vasorelaxant activities with highest R_max_ values of 95% ethanol extracts (72.67 ± 1.61%) > 50% ethanol extracts (73.57 ± 2.99%) > water extracts (55.85 ± 2.35%)	[11]
**(k) Renoprotective Properties**						
Nephroprotective effect of *C. nutans* leaves against cisplatin-induced nephrotoxicity	Aqueous extracts	100, 200 or 400 mg/kg/day for 90 days by oral gavage	Male and female Sprague-Dawley rats (In-vivo)	Cisplatin-induced renal toxicity in rats	Attenuated the renal toxicity and further increase the glomerular filtration rate, serum electrolytes, and urine creatinine excretion	[unpublished data of same lab]
The nephroprotective effect of *C. nutans* in cisplatin-induced nephrotoxicity in the in vitro condition	Water and ethanol extracts	Extractions using different ratios of ethanol to water (0 to 100%, with 20% increment)	Cell viability assay MTT assay, Lactate dehydrogenase (LDH) assay and NMR analysis of cell extract and corresponding culture Media (In-vitro)	Rat renal proximal tubular cells (NRK-52E) line	1000 μg/mL of *C. nutans* leaves extract had the most potential nephroprotective effect against cisplatin toxicity on NRK-52E cell lines at 89% of viability	[64]
Nephroprotective effect of *C. nutans* leaves extract against cisplatin-induced nephrotoxic humankidney cells	Water and ethanol extracts	Extractions using different ratios of ethanol to water (0 to 100%, with 20% increment)	Cell viability assay MTT assay, lactate dehydrogenase (LDH) assay (In-vitro)	human kidney cell (PCS-400–010) culture	Improved the % of cell viability in mitochondrial dehydrogenase activity (MTT) and lactate dehydrogenese (LDH) assay after 24 h pretreatment of the extract	[65]

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
