# Peer review of "A Narrative Review on the Phytochemistry, Pharmacology and Therapeutic Potentials of Clinacanthus nutans (Burm. f.) Lindau Leaves as an Alternative Source of Future Medicine"

_molecules, 2021, doi:10.3390/molecules27010139_

Round 1
Reviewer 1 Report
The effort made by the authors is very valuable, as nowadays it is impossible to follow most of the current literature on a topic of interest. This paper is a comprehensive review focused on the botanical, phytochemistry, and pharmacological properties of Clinacanthus Nutans Leaves. The manuscript fits within the scope of the journal. The manuscript is interesting. The title is clear and it is adequate to the content of the article. The author’s work on discussing achieved results is appreciated. Some revisions are necessary to improve the clarity of the presentation.
I have some recommendations for authors:
Please include some information about the work method: How do you search literature data? How was the period? Which sources?
The photo from Figure 1 is original? If not, please include citations.
Please improve de quality of figure 2 and especially of figure 3. I recommend using a program to write chemical structures.
Review the text for editing. He currently looks messy.
Make a common chapter: ”Conclusions and future perspective”
English language and style are fine but minor spell check is required.
Check the bibliography to be written according to the requirements of the journal.
Author Response
Manuscript number: Molecules-1441488
Title of manuscript: A Narrative Review on the Phytochemistry, Pharmacology and Therapeutic Potentials of Clinacanthus Nutans Leaves as an Alternative Source of Future Medicine
Author acknowledgement:
Corresponding author on the behalf of all co-authors pay gratitude to the editor for generously considering this manuscript for review and authors also thank the reviewer 1 for his positive comments to improve the quality of this manuscript to attract the reader’s community of molecules journal of the MDPI publisher. We have again put our energy to meet the expectations of the editor and reviewer for the publication of this manuscript. However, author welcomes any further suggestion to improve the quality of manuscript.
Comments and Suggestions for Authors
The effort made by the authors is very valuable, as nowadays it is impossible to follow most of the current literature on a topic of interest. This paper is a comprehensive review focused on the botanical, phytochemistry, and pharmacological properties of Clinacanthus Nutans Leaves. The manuscript fits within the scope of the journal. The manuscript is interesting. The title is clear and it is adequate to the content of the article. The author’s work on discussing achieved results is appreciated. Some revisions are necessary to improve the clarity of the presentation.
I have some recommendations for authors:
Comment 1:
Please include some information about the work method: How do you search literature data? How was the period? Which sources?
Response to comment 1:
Suggested modifications related to data search and source of information have been done by incorporating the lines in the abstract part of the updated version of manuscript.
Comment 2:
The photo from Figure 1 is original? If not, please include citations.
Response to comment 2:
Dear reviewer, these photos of plant has been taken directly with the help of professional camera. Model and specifications of the camera has been mentioned in the caption of the image in updated version of the manuscript.
Comment 3:
Please improve de quality of figure 2 and especially of figure 3. I recommend using a program to write chemical structures.
Response to comment 3:
Authors appreciate the comment of reviewer about the quality of the picture. Picture quality is improved in the updated version of the manuscript by using Photoshop for Figure 2 and Chem Draw software for Figure 3.
Comment 4:
Review the text for editing. He currently looks messy.
Response to comment 4:
Manuscript is read and edited by leader of the Team Prof. Emeritus, Edward J. Johns who is well known renal Physiology researcher and native speaker of the English language.
Comment 5:
Make a common chapter:”Conclusions and future perspective”
Response to comment 5:
Modified as suggested by Reviewer 2.
Comment 6:
English language and style are fine but minor spell check is required.
Response to comment 6:
Spelling checks have been rechecked and corrected by spell checking word by word in whole manuscript.
Comment 7:
Check the bibliography to be written according to the requirements of the journal.
Response to comment 7:
We built bibliography by using endnote and some duplicated and retracted papers references have been deleted. Updated version of manuscript contains bibliography as per style recommended by molecule journal of MDPI.
Reviewer 2 Report
The manuscript is a comprehensive account on Phytochemistry, Pharmacology and 2 Therapeutic Potentials of Clinacanthus Nutans. Through out manuscript, Clinacanthus nutans can be maintained as C. nutans.The content of the table must be reduced to a simple form for easy understanding.I think, repetition of content in few tables. Please look at it

Author Response
Manuscript number: Molecules-1441488
Title of manuscript: A Narrative Review on the Phytochemistry, Pharmacology and Therapeutic Potentials of Clinacanthus Nutans Leaves as an Alternative Source of Future Medicine
Author acknowledgement:
Corresponding author on the behalf of all co-authors pay gratitude to the editor for generously considering this manuscript for review and authors also thank the reviewer 2 for his positive comments to improve the quality of this manuscript to attract the reader’s community of molecules journal of the MDPI publisher. We have again put our energy to meet the expectations of the editor and reviewer for the publication of this manuscript. However, author welcomes any further suggestion to improve the quality of manuscript.
Following changes have been done as per suggestion of reviewer 2.
- Name of the plant changed from Clinacanthus nutans to Clinacanthus nutans (Burm. f.) Lindau and abbreviated to C. nutans at 240 places in the manuscript.
- Heading number 2 is changed from ‘’ Phytochemistry and chemical constituents of Clinacanthus nutans leaves’’ to ‘’ Phytochemistry’’ only as suggested by reviewer.
- Structure of phytoconstituents have been replaced by using Chemdraw software in updated version of manuscript.
- Reviewer 2 suggested to minimize the information in the Table 6. Authors appreciate the suggestions of reviewer 2 but at the same time reducing the information in Table 6 will lose the elegance of information provided in this narrative review. However, upon reviewer 2 and editor suggestions information can be minimized.
Although all changes have been done as suggested by the reviewer 2 but any necessary changes in manuscript can be done.

Round 2
Reviewer 1 Report
The authors improved the manuscript and made the requested changes.